

# Roots induce hydraulic redistribution to promote nutrient uptake and nutrient cycling in nutrient-rich but dry near-surface layers

Jing Yan[1],[*] and Teamrat A. Ghezzehei[2],[*]

[1]Plant & Soil Sciences, University of Delaware
[2]Life & Environmental Sciences, University of California, Merced

**Correspondence:** Jing Yan (yanjing@udel.edu)

**Abstract.** The rhizosphere is an exclusive passage for water and nutrients from the bulk soil to the whole plant. As such, its importance significantly outweighs the limited volume it represents. Numerous recent experimental and modeling studies have shown that plants invest considerable resources in modifying this region. However, roots must also balance the significant differences in resource availability in the vast soil volume they inhabit. Studies suggest that hydraulic redistribution by roots

helps counterbalance the large differences in water status experienced by roots. In addition, experimental evidence suggests that hydraulic redistribution plays a role in mitigating drought effects and aiding nutrient uptake. However, whether hydraulic redistribution is a passive happy accident or a process controlled by plants remains unclear. Here, we present a novel mathematical model that integrates rhizosphere-scale modification of soil hydraulic properties by root exudation with long-distance interaction between roots that occupy disparately resourced soil regions. The model reproduces several known phenomena.

First, hydraulic redistribution is proportional to the hydraulic gradient between wet and dry regions. Its magnitude substantially increases with the accumulation of hydrophilic rhizodeposits in the rhizosphere of the dry region. However, its effect on net water uptake by the whole root system is meager, negating the current hypotheses that hydraulic redistribution helps mitigate drought. Second, hydraulic redistribution facilitates nutrient uptake. We observed that periodic rewetting of nutrient-rich but dry soil layers significantly increases the active uptake of soluble nutrients. Moreover, cyclic rewetting of the rhizosphere

increases the mineralization of the organic matter, thereby releasing nutrients locked in soil organic matter. The latter is another mechanism that supports the well-known phenomenon of priming of organic matter mineralization by root exudation. Overall, our model supports a hypothesis that roots faced with nutrient and organic matter accumulation in unfavorably dry soil regions facilitate hydraulic redistribution via exudation and benefit from the increased nutrient uptake and mineralization rate. These mechanisms could crucial role in determining whether plants can adapt to shifts in resource distributions under a changing

climate.

## 1 Introduction

Plant roots often inhabit soils with patchy and mismatched water and nutrient distributions (Yan et al., 2020; Prieto et al., 2012b). They adapt to these environments by employing strategies that involve short-range rhizospheric soil engineering and long-range matching of root architecture to the unique resource distributions they face (Ahmed et al., 2018). Therefore, the





phenotypic plasticity that enables these strategies (Sultan, 2003) is crucial in determining the fate of plants in a rapidly changing climate. In this paper, we provide a conceptual and mathematical basis for a proposed mechanism that allows plants to exploit nutrient-rich but dry shallow soils by transferring water from wetter deep layers. This study builds up on separate advances in our understanding of how roots modify their immediate surrounding and hydraulic redistribution.

Roots invest a considerable proportion of their photosynthate in altering the rhizosphere–an exclusive window through
which they interact with the soil that surrounds them (Nguyen, 2003; Kuzyakov and Razavi, 2019). High-resolution imaging studies and numerical simulations of water dynamics in the rhizosphere have shown that the accumulation of hydrophilic rhizodeposits substantially increases wetness of rhizosphere and plant-water uptake (Ahmed et al., 2014, 2018; Carminati et al., 2010, 2011; Ghezzehei and Albalasmeh, 2015). Carminati et al. (2016) argued that the rhizosphere's carbon investment pays for itself through increased water uptake and photosynthesis. Roots also exude compounds that facilitate nutrient uptake
(e.g., phosphate) and prime biogeochemical cycles (Spohn and Kuzyakov, 2014; Razavi et al., 2016). The architecture and function of roots must also conform to the peculiar resource distributions and environmental conditions they encounter at the scale of the entire root system. Therefor, plant adaptation to heterogeneous soils calls for intricate and synchronized actions by roots that inhabit distant and disconnected parts of the soil profile. Hydraulic redistribution (HR), the transfer of water from wet to dry parts of the soil via roots, is a prime example of such long-range interactions.

It is well established that a substantial difference in water potential between root-occupied parts of the soil is a necessary condition for HR (Caldwell and Richards, 1989; Neumann and Cardon, 2012; Amenu and Kumar, 2008; Kitajima et al., 2013; Prieto et al., 2012a; Bogie et al., 2018). Nevertheless, it remains unclear whether HR is an unavoidable spontaneous response to a water-potential gradient or a process that requires facilitation by plants. We recently reported that lab-grown tomatoes are more likely to induce HR if the dry (receiving) part of the soil contained plant-available nutrients (Yan et al., 2020). Similarly,
Cardon et al. (2013) and Prieto et al. (2012b) observed that the magnitude of HR performed by sagebrush and Boiss shrubs increased in the presence of nitrogen. The exact mechanism by which roots can induce HR is, however, not known.

Here, we present a modeling study that demonstrates that alteration of rhizosphere soil by rhizodeposition facilitates HR. Moreover, we show that the magnitude of HR has a positive influence on the active uptake of plant-available nutrients and the rate of organic matter mineralization. This study sheds light on how plants can adapt to non-ideal resource distribution
through localized resource investments and long-range synchronized interaction between roots that inhabit variably resourced soil regions (Schnepf et al., 2022).

## 2 Mathematical Model

### 2.1 Conceptual Model

The conceptual basis of the numerical model is schematically illustrated in Figure 1. While a sufficient amount of soil water
to meet the evapotranspiration demand is available in the deep layers, the plants must rely on the dry, nutrient-rich shallow soils for their nutrient requirements. In this regard, we postulate crucial HR functions that do not appear to have been fully appreciated hitherto: HR water enables nutrient extraction by mobilizing nutrients and promoting microbial mineralization.



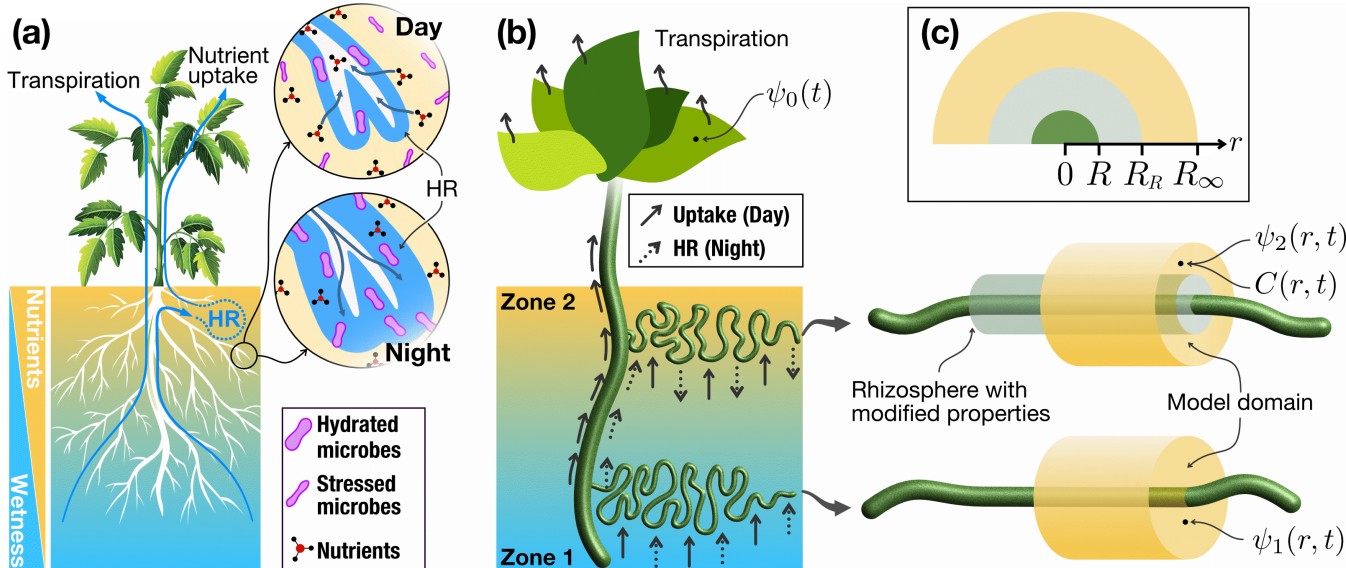

**Figure 1.** Schematic representation of (a) how HR supports nutrient uptakes in dry nutrient-rich shallow layers across scales. During the day, plant transpiration in deep layers supports a major fraction of water demands, while water depletion in shallow layers carries nutrients for nutrient uptake. During the night, HR transfers water from the deeper layer to shallow layers that enables nutrient extraction by mobilizing nutrients and promoting microbial mineralization; (b) mathematical models that incorporate rhizodeposits. The whole root system is conceptually represented by two long connected tubes of R that inhabit the deep (zone 1) and shallow (zone 2) layers of the soil profile. Transpiration and nutrient uptake during the daytime was represented by solid arrows, while HR water transfer was represented by dashed arrows. Dynamics in water and nutrient uptake were modeled in cylindrical domains of radius $R_\infty$ outside the root tubes. Regulation of rhizodeposits was incorporated within the cylindrical domain of radius $R_R$ in zone 2; (c) a cross-section of the model domain: illustrating the root radius ($R$), the outer boundary of rhizodeposition ($R_R$), and the outer boundary of the model domain ($R_\infty$).

More importantly, we hypothesize that plant roots can actively facilitate HR by releasing rhizodeposits that alter their immediate surroundings' hydraulic characteristics.

This study introduces a mathematical model that enables us to test this hypothesis and quantitatively compare the relative benefits of rhizodeposition on the magnitude of active nutrient uptake and mineralization rate. These questions call for a multi-scale model that seamlessly represents both the short-range flow and transport in the rhizosphere as well as the long-range interactions between different parts of the root system. The whole root system is conceptually represented by two long connected tubes that inhabit the deep (zone 1) and shallow (zone 2) layers of the soil profile as shown in Figure 1b. Each of

these two roots is surrounded by a shell of rhizosphere and bulk soil, where water flow and nutrient transport are mediated by plant uptake. The roots in each layer are assumed to be of uniform radii ($R$) and hydraulic characteristics, such that a single cross-section, orthogonal to the axis of the root, is sufficient to represent the flow and transport processes.



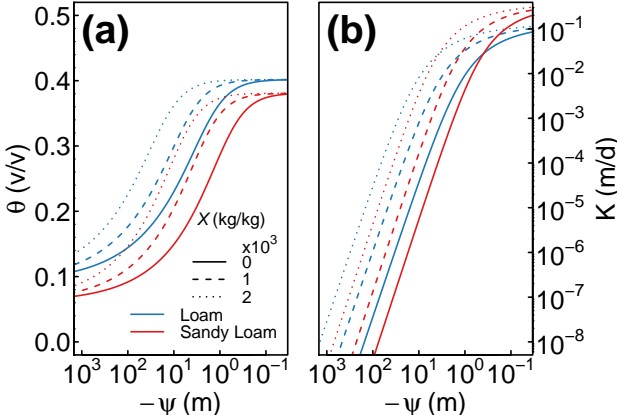

**Figure 2.** Hydraulic properties of of the two example soils used in this study. Solid lines represent typical Sandy Loam and Loam soils, with hydraulic parameters derived from ROSSETTA pedo-transfer function (Zhang and Schaap, 2019). Dashed and dotted lines represent modifications due to two levels of rhizodeposit concentrations (Ghezzehei and Albalsmeh, 2013).

To enable accurate accounting of mass balance, the model domain is defined by an imaginary, non-overlapping concentric shell of soil as illustrated by the yellow cylinder in Figure 1b and 1c. The radius of this shell is given by,

$$R_\infty = \frac{1}{\sqrt{\pi \rho_R}} \tag{1}$$

where $\rho_R$ [L L$^{-3}$] is the root-length density, which represents only the portion of the root system that is actively involved in water and nutrient uptake. Therefore $\rho_R$ denotes only a fraction of the total root-length density. The specific-surface area ($A$ [L$^2$L$^{-3}$]) of the soil-root interface in each zone is given by

$$A = 2\pi R \rho_R \tag{2}$$

Moreover, we also define another inner shell illustrated by light-green and denoted by $R_R$ in Figure 1b, which represents the region where most of the rhizodeposition is accumulated.

In the following, we consider two distinct mechanisms by which rhizodeposition-facilitated HR can enhance nutrient acquisition from relatively dry soil patches. First, in §2.2–2.4, we present the role of HR on the active root uptake of plant-available non-sorbing, non-reactive nutrients. Secondly, in §2.5, we consider the effect of elevation of soil wetness by HR on the rate of organic matter mineralization.

## 2.2 Flow and Transport Equations

The soil hydraulic functions are described using Mualem-van Genuchten model (van Genuchten, 1980)

$$\Theta_i = (1 + (\alpha \psi_i)^n)^{-m} \tag{3a}$$





$$K_i = K_s \Theta_i^{1/2} \left( 1 - (1 - \Theta_i^{1/m})^m \right)^2 \tag{3b}$$

where $i \in \{1, 2\}$ denotes zone 1 and zone 2; $\Theta_i = (\theta_i - \theta_r)/(\theta_s - \theta_r)$ is effective saturation, with $\theta_s$ and $\theta_r$ representing the saturated and residual water content; and $\alpha, n, m = 1 - 1/n$ are shape parameters. In this study we use the hydraulic parameters of typical loam and sandy loam soils derived using the ROSSETTA pedotransfer function (Zhang and Schaap, 2019), which are depicted by solid lines in Figure 2 (see parameters in Table 1).

Within the soil domain bounded by $R \leq r \leq R_\infty$, we consider water flow in both zones and nutrient transport in zone 2. These are described by coupled governing partial differential equations, which can be written in a generalized form as

$$\mathbf{c} \frac{\partial \mathbf{u}}{\partial t} = r^{-1} \frac{\partial}{\partial r} (r\mathbf{f}) \tag{4}$$

where $\mathbf{u}$ represents the three independent variables:

$$\mathbf{u} = [\psi_1, \psi_2, N] \tag{5}$$

The first two are matric heads in zone 1 and zone 2, while the third is dimensionless nutrient concentration in the soil volume defined as

$$N = \frac{N_l \theta_2}{N_{\max}} \tag{6}$$

where $N_l$ is nutrient solution concentration in pore-water and $N_{\max}$ is the maximum total nutrient concentration per unit soil volume.

The coefficient $\mathbf{c}$ denotes the hydraulic capacity functions for the water flow equations and it is unity for the transport equation of nutrient flow,

$$\mathbf{c} = \left[ \frac{d\theta_1}{d\psi_1}, \frac{d\theta_2}{d\psi_2}, 1 \right] \tag{7}$$

The first two elements of the coefficient $\mathbf{f}$, $f_1$ and $f_2$, represent the Buckingham-Darcy law for the water flow equations. The third element ($f_3$) is non-reactive advection-diffusion equation for the nutrient transport equation

$$\mathbf{f} = \left[ K_1 \frac{\partial \psi_1}{\partial r}, K_2 \frac{\partial \psi_2}{\partial r}, D_e \frac{\partial N}{\partial r} + \frac{K_2}{\theta_2} \frac{\partial \psi_1}{\partial r} N \right] \tag{8}$$

The effective diffusion-dispersion coefficient $D_e$ varies with water flux density and water content and is given by

$$D_e = \lambda \frac{K_2}{\theta_2} \frac{\partial \psi_2}{\partial r} + D_0 \frac{\theta_2^{10/3}}{\theta_s^2} \tag{9}$$

where $\lambda$ is dispersivity and $D_0$ the diffusivity of the nutrient in free water.



**Table 1.** Soil hydraulic parameters

| Soil Texture | $\theta_s$ | $\theta_r$ | $n$ | $\alpha$ [m$^{-1}$] | $K_S$ [m s$^{-1}$] |
|---|---|---|---|---|---|
| Loam | 0.5 | 0.1 | 2 | 0.001 | 10 |
| Sandy Loam | 0.5 | 0.1 | 3 | 0.001 | 10 |

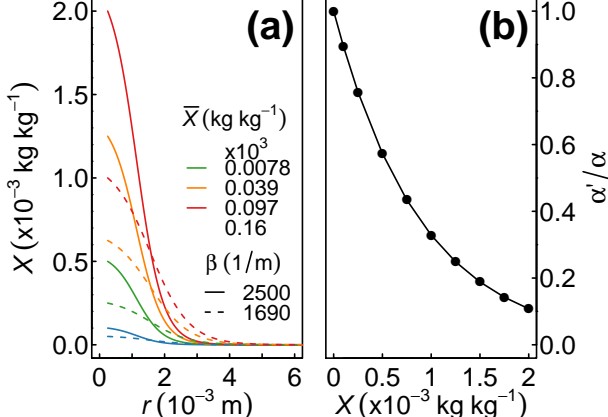

**Figure 3.** Selected example of rhizodeposit distributions. Solid and dashed lines represent narrow and wide spreading of the organic molecules, respectively. Colors denote levels of total rhizodeposit concentrations in the $r \leq 2.5 \times 10^{-3}$ m region.

## 2.3 Rhizosphere Modifications

The spatial distribution of the rhizodeposit mass fraction is described by (Ghezzehei and Albalasmeh, 2015)

$$X = \frac{\xi X_0}{1 + (\xi - 1)e^{\beta(r-R)}} \quad (10)$$

where $X_0$ [kg kg$^{-3}$] is the mass fraction at the root-soil interface $\beta$ [m$^{-1}$] is shape factor that describes the extent of rhizodeposition spread, and $\xi$ is a shape factor that describes the rate of fall near the root surface. The latter was set at $\xi \approx 1.11$ as explained in Appendix A. The average rhizodeposit concentration in the rhizosphere is given by

$$\bar{X} = \frac{\int_R^\infty X dr}{\pi(R_R - R)} \quad (11)$$

where $R_R$ is the outer extent of the rhizodeposit spread, which also marks the boundary of the altered rhizosphere. Evaluation of the integral in Eq (11) is detailed in Appendix A. In this study we consider 10 different levels of average rhizodeposit concentrations, excluding a rhizodeposit free control. In addition, for each level of rhizodeposition we varied the values of the $X_0$ and $\beta$ parameter to achieve narrow and wide distributions. Specifically, the value of $X_0$ of the narrow distribution was

twice that of the wide distribution. The corresponding spread shape factors were set at $\beta = 2500$ m$^{-1}$ and $\beta = 1690$ m$^{-1}$ for



**Table 2.** Parameters of rhizodeposition used for simulations.

| $\bar{X}\,(\times 10^{-3}\,\mathrm{kg\,kg^{-1}})$ | Narrow | | Wide | |
| --- | --- | --- | --- | --- |
| | $X_0\,(\times 10^{-3}\,\mathrm{kg\,kg^{-1}})$ | $\beta$ | $X_0\,(\times 10^{-3}\,\mathrm{kg\,kg^{-1}})$ | $\beta$ |
| 0.0000 | 0.000 | 2500 | 0.000 | 1690 |
| 0.0078 | 0.100 | 2500 | 0.050 | 1690 |
| 0.019 | 0.250 | 2500 | 0.125 | 1690 |
| 0.039 | 0.500 | 2500 | 0.250 | 1690 |
| 0.058 | 0.750 | 2500 | 0.375 | 1690 |
| 0.078 | 1.000 | 2500 | 0.500 | 1690 |
| 0.097 | 1.250 | 2500 | 0.625 | 1690 |
| 0.12 | 1.500 | 2500 | 0.750 | 1690 |
| 0.14 | 1.750 | 2500 | 0.875 | 1690 |
| 0.16 | 2.00 | 2500 | 1.000 | 1690 |

the narrow and wide distributions, respectively. A selection of wide and narrow rhizodeposition distributions at four levels are shown in Figure 3a.

In the remainder of this paper, the nominal definition of the rhizosphere is the region of the soil within 10 times the radius of the root ($10 \times R$). The boundary of the rhizosphere is marked with a dotted line in Figure 3a, which coincides with outermost extent of the narrow distribution of rhizodeposits.

The water retention characteristic of the rhizosphere is locally modified in proportion to the mass fraction of rhizodeposition (Ghezzehei and Albalasmeh, 2015)

$$\alpha' = \alpha e^{aX} \tag{12}$$

where $a$ a fitting parameter determined experimentally using synthetic mucilage. The relative effect of rhizodeposition on the $\alpha$ parameter is illustrated in Figure 3b.

## 2.4 Boundary and Initial Conditions

The connectivity between the two root zones and with the transpiring leaves is incorporated as flux boundary condition at the inner boundary, $r = R$. An important aspect of this is the day-night cycle, which is represented by smoothed step function that switches between 0 (closed stomata) and 1 (open stomata) every 12 hours:

$$w = \frac{1}{2} + \frac{\sin(2\pi\tau)}{2}\left(\frac{1+b}{1+b\sin(2\pi\tau)^2}\right)^{1/2} \tag{13}$$

where $\tau$ is time in days and $b$ is a parameter that determines the smoothness of the transition. The leaf water potential $\psi_0$ is set to prescribed value of $\psi_0^*$ during day time. At night, it is equilibrated with the soil water potential at the root surface ($r = R$) in





zone 1

$$\psi_0 = \psi_0^* w + (1-w)\psi_1|_{r=R} \tag{14}$$

Then the water flow boundary conditions at $r = R_b$ are expressed by

$$f_1|_{r=R_b} = \{\psi_1 - \psi_0 - (1-w)(\psi_2 - \psi_0)\}\frac{\kappa}{A} \tag{15a}$$

$$f_2|_{r=R_b} = \{\psi_2 - \psi_0\}\frac{\kappa}{A} \tag{15b}$$

where $\kappa$ is the effective conductance that represents of the soil-plant-atmosphere interface (Sperry et al., 1998). The total
transpiration flux from zones 1 and 2 is given by

$$T = T_1 + T_2 = A\left(f_1|_{r=R} + f_2|_{r=R}\right) \tag{16}$$

The above equations ensure that transpiration occurs ($T > 0$) during the day ($w = 1$), whereas HR (flow from zone 1 to zone
2) occurs ($T_1 = -T_2$) at night ($w = 0$), provided that there is sufficient potential gradient to overcome gravitational potential
difference between the two zones.

We consider only active nutrient uptake, which may be described using Hill-Langmuir equation. This equation is a variation
of Michaelis-Menten model used by (Espeleta et al., 2017) and has an additional coefficient ($h$) that characterizes the sensitivity
of the response function,

$$f_3|_{r=R_b} = F_{\max}\frac{N'^h}{N'^h + K_N^h} \tag{17}$$

where $N' = N/\theta_2 = N_l/N_{\max}$ is the dimensionless nutrient concentration in the pore water, $K_N$ is the half-saturation concen-
tration of nutrients, and $F_{\max}$ is the maximum uptake rate. For $h > 1$ the function exhibits a sigmoidal shape, which represents
a threshold effect at ultra low concentrations. The nutrient transport parameters used in this study correspond to that of nitrate
and were derived from a similar modeling study of the rhizosphere (Espeleta et al., 2017) (see Table 2)

Recall that the volume of the model domain is bounded by an imaginary non-overlapping outer surface at $r = R_\infty$. This
assumption prevents transfer of mass accross the the outer boundary and is represented by zero-flux of water and nutrients

$$\mathbf{f}|_{r=R_\infty} = 0 \tag{18}$$

For all the simulations we performed the subsurface layer (zone 1) was assumed to be initially at field capacity, $\psi_1(t = 0) = -3$ m. Several scenarios were simulated by considering a wide range of initial matric head values in zone 2, $-100$ m $\leq \psi_2(0) \leq -10$ m. The initial dimensionless nutrient concentration is set to unity.

Volume-weighed mean water content ($\bar{\theta}_2$) and mean matric potential ($\bar{\psi}_2$) are calculated as

$$\bar{\theta}_2 = \frac{1}{R_*^2 - R^2}\int\limits_R^{R_*} r\theta_2 dr \tag{19a}$$





$$\bar{\psi}_2 = \frac{1}{R_*^2 - R^2} \int\limits_R^{R_*} r\psi_2 dr \tag{19b}$$

where $R_*$ is the outer extent of the volume of interest, which may be set at the outer extent of the rhizosphere ($R_R$) or the model boundary ($R_\infty$). The latter, $R_* = R_\infty$, results in the mean values of the entire zone 2.

## 2.5 Nutrient Mineralization Potential

An important consequence of alteration of the rhizosphere water content by HR is acceleration of rhizospheric C mineralization and nutrient cycling. This can be an important mechanism for plants to facilitate nutrient availability in water-limited systems. For simplicity, we assume a single pool of C and first-order decay dynamics

$$\frac{dC}{dt} = -kC \tag{20}$$

where $C$ is dimensionless organic C concentration and $k = k_o k_w \, [\mathrm{T}^{-1}]$ is the effective decay constant that accounts for decay rate at optimal physical environmental condition ($\kappa_o, [\mathrm{T}^{-1}]$) and dimensionless sensitivity function ($k_w$), which represents sensitivity to moisture and matric head (Ghezzehei et al., 2019)

$$k_w = e^{s\psi_2} \frac{\theta_2^{10/3}}{\theta_s^2} \tag{21}$$

where $s \, [\mathrm{m}^{-1}]$ is an empirical coefficient. We ignored the limiting effect of low aeration in Eq (21) because the focus of this study is on organic-matter accumulation in the moisture-limited zone 2. Equation (21) describes the diurnal cycles of hydration and stress experienced by the rhizospheric microbes, as illustrated in Figure 1a. Therefore, this model is deemed effective in describing the role of rhizodeposition in increasing nutrient availability by accelerating organic matter mineralization.

## 3 Results

### 3.1 Spatial and Temporal Patterns of Water and Nutrient Dynamics

Representative simulations that illustrate the effect of rhizodeposition on matric head dynamics under identical initial and boundary conditions are presented in Figure 4. The left column is for rhizodeposit-free soil (homogeneous hydraulic characteristics), whereas the right column is for the rhizosphere soil altered by narrowly distributed rhizodeposition of $\bar{X} = 0.097 \times 10^{-3}$ kg kg$^{-1}$.s The leaf water potential ($\psi_0$) and the mean matric heads of the soils in zone 1 ($\bar{\psi}_1$), and zone 2 ($\bar{\psi}_2$) are shown in the top row of Figure 4. Both scenarios exhibited HR, as indicated by the mean matric head's diurnal fluctuation in the dry zone 2. However, rhizodeposit accumulation allowed the rhizosphere to retain more water at the same matric head, resulting in faster water flux. As a result, the HR signal appears to be more pronounced in the presence of rhizodeposition. Moreover, the extent of the rhizosphere that experiences drying and rewetting is wider, as shown in the bottom row of



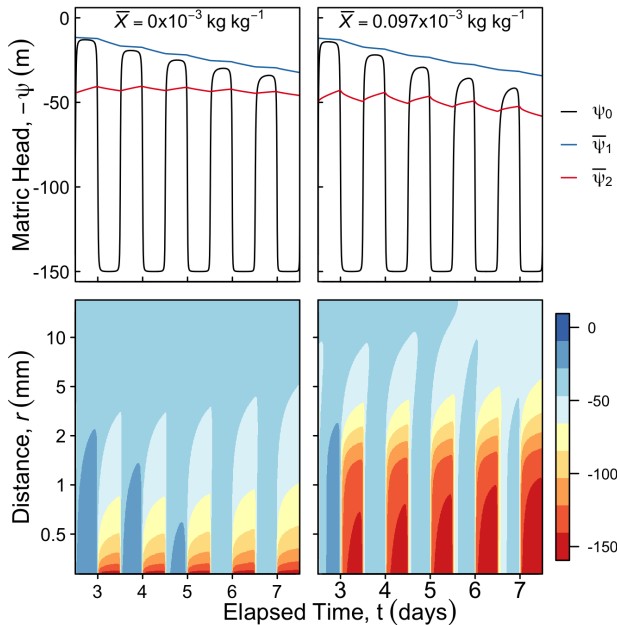

**Figure 4.** Typical temporal and spatial dynamics of matric head for the Sandy Loam soil without (left column) and with (right column) narrow distribution of rhizodeposition. The top row shows the leaf water potential ($\psi_0$) and mean matric heads of zone 1 ($\psi_1$) and zone 2 ($\psi_2$). The bottom row shows the spatio-temporal dynamics of matric head (color indicates matric head).

Figure 4. In the absence of rhizodeposition, the matric head remains largely unaffected beyond 1 mm from the center of the root. Whereas in the presence of rhizodeposits, significant drying and rewetting were observed as far as 5 mm from the center of the root, which accounts for $> 30$ times larger soil volume than the rhizodeposit free soil.

In Figures 5–7, we present temporal and spatial variations of matric head and nitrate concentration for one level of initial matric head in zone 2 ($\psi_2 = -50$ m) and three levels of rhizodeposits (none, intermediate and maximum), which are consistently color coded. The arrangement of the the subplots is consistently repeated in all subsequent figures. The Sandy Loam (SL) and Loam (L) soils are reported in the top and bottom rows, respectively. The narrow and wide distributions of the rhizodeposits are shown on the left and right columns, respectively.

In Figure 5, we show the matric head dynamics in zone 2 over the course of ten days. The occurrence of HR is clearly evident for the selected three levels of rhizodeposits (none, medium, highest) shown. The degree of drying that occurred during day time (indicated by minimum mid-day matric head) increased with the amount of rhizodeposits for all four cases. In contrast, little difference was observed in the degree of nocturnal maximum level of matric head, which appears to be primarily controlled by the matric head in zone 1.

Notice that the magnitude of HR reflected within the rhizosphere is more pronounced than the mean for the bulk soil (c.f. Figure 4) because the HR water is mostly localized with close proximity of the roots. The spatial distribution of HR

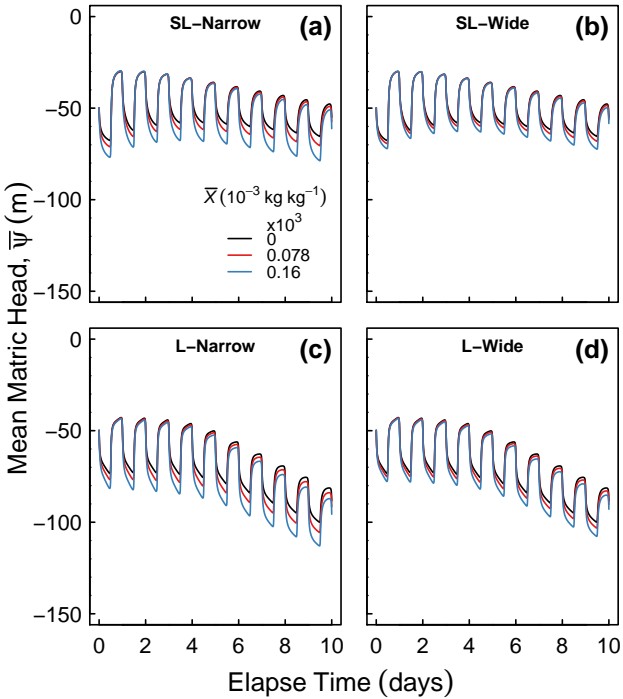

**Figure 5.** Temporal variations of matric head for one level of initial matric head in zone 2 ($\psi_2 = -50$ m) at three levels of rhizodeposits (none, intermediate and maximum). Left column (a and b) are for narrow rhizodeposit distributions and right (c and d) are for wide distributions. Top row (a and c) are for sandy loam soil and bottom row (b and d) is for loam soil.

water is further illustrated in Figure 6, where matric head at noon and midnight on the seventh day are depicted. The water content distribution is proportional to the rhizosphere concentration in the vicinity of the rhizosphere. Because elevated wetness

increases the hydraulic conductivity, over time the soils with higher rhizodeposition experience more net water uptake. At farther distance from the roots, the water content is inversely proportional to the rhizodeposition level.

Furthermore, elevated water content results in more pronounced bidirectional transport and active uptake of nitrate. The resulting complex interplay between the two processes is illustrated in Figure 7, which shows spatial nitrate distributions at noon and midnight, on the seventh day. The displayed total nitrate concentration is expressed on the basis of the soil volume. In

all cases, the nitrate concentration exhibits back and forth motion as water is taken up and released by the roots. As in the case of the water content, much of the dynamics in localized close to the roots. The net nutrient uptake is noticeable as a drop in the nitrate concentration at the far end of the soil volume. Generally, nutrient uptake increases with rhizodeposit concentration and the Loam soil appears to result in more nitrate uptake than the Sandy Loam soil. There is no discernible difference between narrow and wide distributions.


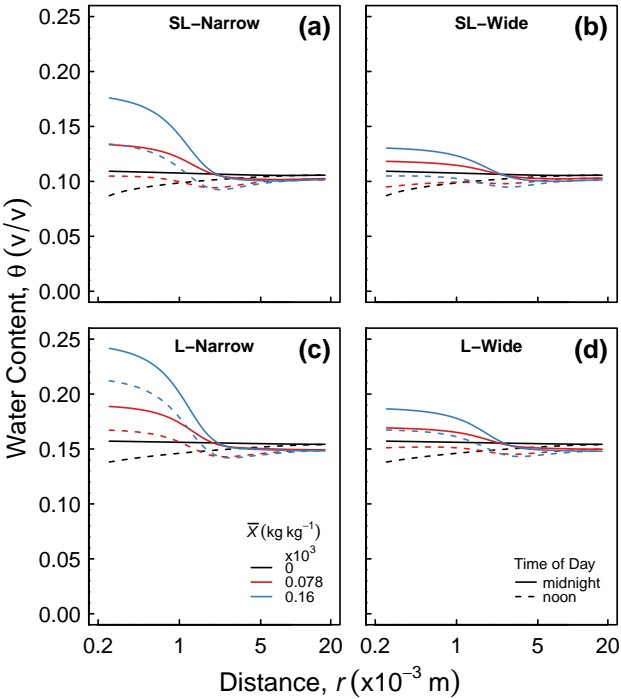

**Figure 6.** Spatial variations of matric head at noon and midnight on the seventh day for one level of initial matric head in zone 2 ($\psi_2 = -50$ m) at three levels of rhizodeposits (none, intermediate and maximum). Left column (a and b) are for narrow rhizodeposit distributions and right (c and d) are for wide distributions. Top row (a and c) are for sandy loam soil and bottom row (b and d) is for loam soil.

## 3.2 Effects on Net Water and Nitrate Uptake

In Figures 9 and 10, we report the net water and nutrient uptakes, respectively, over the course of ten days. Because the actual values of uptake vary by a large magin across range of wetness levels we explores, we present the results as percent differences from the rhizosphere soil. The lines represent five different levels of initial matric in zone 2, $\psi_2 \in \{-10, -30, -50, -70, -100\}$ m. For easy identification, the highest and lowest values are indicated by up-facing and down-facing triangles, respectively, and the intermediate value is shaded solid color. In addition, the colors transition from cold (blue) at the wettest end to warm (red) at the driest end. For all levels of initial wetness of zone 2, water and nutrient uptake increased with the quantity of rhizodeposition. But both water and nutrient uptakes exhibited notable differences in trends between soil types, wetness levels.

The incremental effect of rhizodeposition (slope) on net water uptake decreased with increasing rhizodeposit level for both soils, with the narrow distribution showing more pronounced decline than the wide distribution. For all levels of rhizodeposition, the net water uptake progressively increased with initial wetness for the Sandy Loam soil (Figure 9a and 9b). In contrast, the effect of initial matric head was less pronounced and non-linear for the Loam soil (Figure 9c and 9d), with the highest level



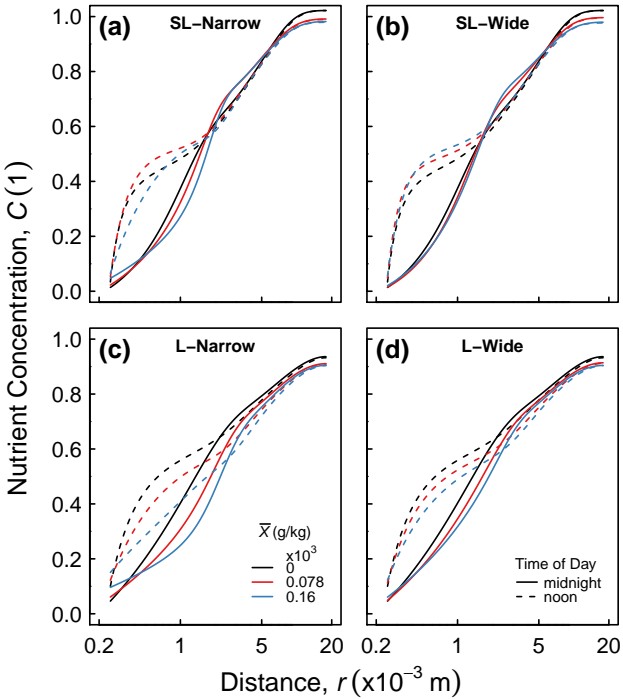

**Figure 7.** Effect of elevated rhizosphere wetness on daytime and nighttime relative nutrient concentrations for various rhizodeposit concentrations. Left column (a and b) are for narrow rhizodeposit distributions and right (c and d) are for wide distributions. Top row (a and c) are for sandy loam soil and bottom row (b and d) is for loam soil.

of increase observed $\psi_2 = -30$ m. Overall, the net effect on water uptake is very low (less than 6 %) because the wetter zone 1 provides the bulk of the transpired water.

In contrast, rhizodeposits resulted in up to 150 % in net nitrate uptake increase. The effect is more three times more pro-
nounced in the Sandy Loam soil than the Loam soil. Unlike the effect on water uptake, net increase in nutrient uptake increased with decrease in the initial matric head of zone 2. However, for the Sandy Loam soil, decreasing the initial matric head below $\psi_2 = -70$ m resulted in decrease in net nutrient uptake as well. For both soils, the narrow rhizodeposit distribution resulted in modest gain of net nitrate uptake compared to the wider distribution.

### 3.3 Effect on Net Nutrient Mineralization

Assuming the SOM reserve is not significantly depleted within the short period (10 days), we can obtain the total mineralized C in the rhizosphere by

$$C_T = \int\limits_{R}^{R_r} \int\limits_{0}^{T} r k_w(r,t)\, dt\, dr \tag{22}$$

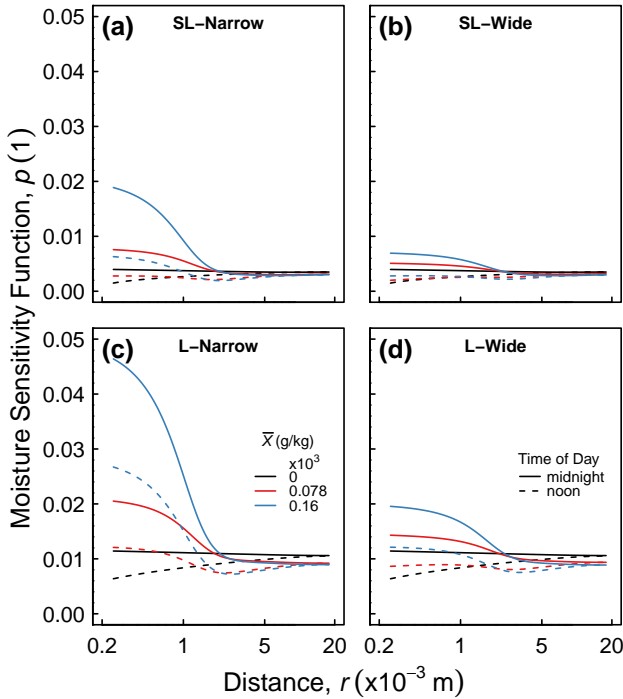

**Figure 8.** Effect of elevated rhizosphere wetness on daytime and nighttime relative mineralization rate for various rhizodeposit concentrations. Left column (a and b) are for narrow rhizodeposit distributions and right (c and d) are for wide distributions. Top row (a and c) are for sandy loam soil and bottom row (b and d) is for loam soil.

where $T = 10$ days. In Figure 11, we show the effects of HR on the increase in net mineralization over a simulation period of ten days. Similar to net water and nutrient uptake, increase in net mineralization depends on both the soil types and initial wetness levels. Besides,a narrower distribution of rhizodeposits further promote the net mineralization, in contrast, such effect is less pounced for wider distribution of rhizodeposits.

The incremental effects of rhizodeposits on net mineralization consistently increased with increasing rhizodeposit levels for Loam soil. Similar effects occurred under lower wetness levels for silt Loam soil, in contrast, under higher wetness levels, net mineralization varied with rhizodeposit levels. Specifically, at the lower rhizodeposit levels (i.e., $< 0.10 \times 10^{-3}$ kg kg$^{-1}$), net change in mineralization decreased and became negative, while at higher rhizodeposit levels it increased exponentially up to nearly $40\%$.

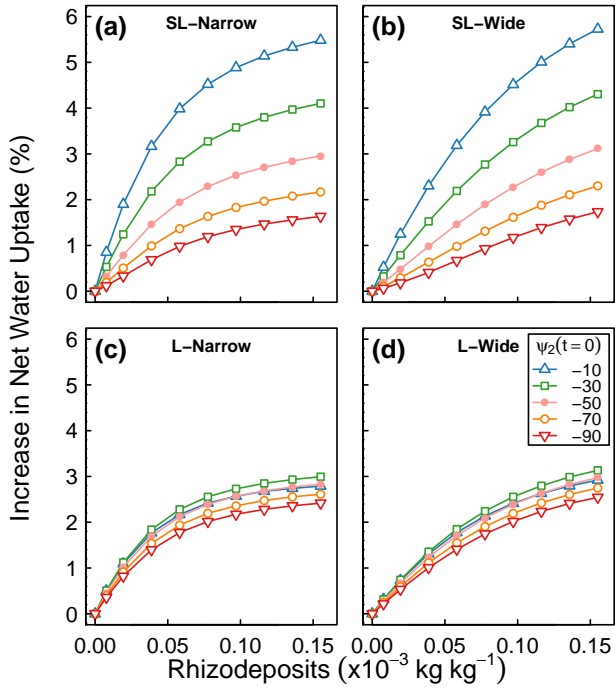

**Figure 9.** Effect of elevated rhizosphere wetness on relative transport and passive uptake of nutrients from the effective soil volume occupied by roots ($0.25$ mm $\leq r \leq 17.5$ mm) for various initial matric of the dry zone ($\psi_2(t=0)$). Left column (a and b) are for narrow rhizodeposit distributions and right (c and d) are for wide distributions. Top row (a and c) are for sandy loam soil and bottom row (b and d) is for loam soil.

# 4 Discussion

## 4.1 Effects of Rhizodeposits on HR pattern and Magnitude

This work indicates that for the simulated scenarios of various soil texture and dryness, the presence of rhizodeposits by roots

was found to affect both patterns and amounts of HR. We assumed identical root length and conductance and thus effects of rhizodeposits on HR were identified. The higher amounts of rhizodeposits increased soil water content due to HR-induced wetting, especially in the immediate vicinity of root surfaces. This increase resulted from the higher water holding capacity and elevated unsaturated hydraulic conductivity in soils with rhizodeposits, compared to rhizodeposit-free soils, as demonstrated in Carminati et al. (2011); Ghezzehei and Albalasmeh (2015). The elevated water content enabled a higher hydraulic conductivity

under drier conditions and extended the radial footprint of wetting and drying from HR and water uptake by roots. Even with smaller amounts of rhizodeposits, $< 0.1 \times^{-3}$ kg kg$^{-1}$, we observed a more than two fold increase in soil water content and up to 30 times increase in soil volumes for water and nutrient extraction (Figure 4, bottom and 6). This result is consistent with

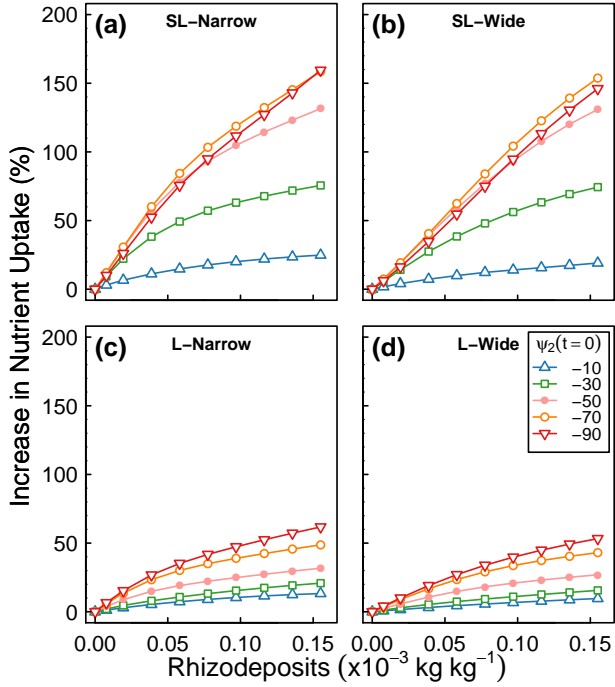

**Figure 10.** Effect of elevated rhizosphere wetness on relative rate of mineralization in the immediate region around roots ($0.25$ mm $\leq r \leq$ $2.5$ mm) for various initial matric of the dry zone ($\psi_2(t=0)$). Left column (a and b) are for narrow rhizodeposit distributions and right (c and d) are for wide distributions. Top row (a and c) are for sandy loam soil and bottom row (b and d) is for loam soil.

previous reports of sensitivity analysis where an increase in radial root-soil conductance enhanced HR magnitude by factors of 140 to 200 % (Wang, 2011).

The spatial distribution of rhizodeposits appears to be another important factor in enhanced HR wetting. A narrower rhizodeposit distribution apparently further enhanced HR magnitude, as reflected by higher soil water content in the proximity of roots at the midnight. For both soils, the narrow rhizodeposit distribution also resulted in modest gain of net water and nitrate uptake compared to the wider distribution, especially at the lower to moderate rhizodeposit levels. The simulated benefit of narrow rhizodeposit distribution may result from its higher ability in sustaining soil hydraulic conductivity that favor HR wet-

ting, and hence water and nutrient uptake under pronounced drying conditions, compared to wider rhizodeposit distribution. A higher soil water content at the close proximity of the root in Zone 2 with narrow rhizodeposit distribution may progressively increase unsaturated hydraulic conductivity, especially during day time when low water potentials occurred (Figure 6). This result further confirmed that typically low diffusivity of rhizodeposits localized within a narrow plane surrounding root tips may have additional benefits, as hypothesized by Ghezzehei and Albalasmeh (2015).

Increasing coarser components of soil in the simulations, i.e., sandy loam versus loam soil, increased the benefit of rhizodeposits for water and nutrient uptake for all levels and distribution of redeposits. Coarser soils tend to drain more easily than



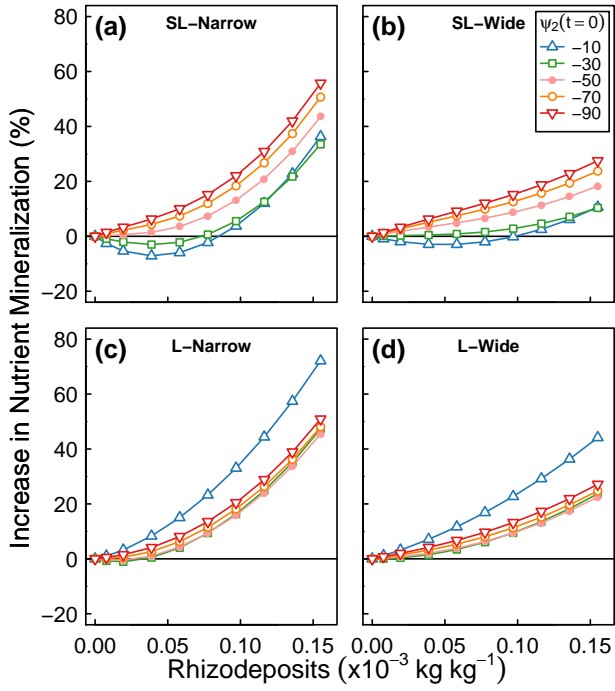

**Figure 11.** Effect of elevated rhizosphere wetness on relative rate of mineralization in the immediate region around roots ($0.25 \times 10^{-3}$ m $\leq r \leq 2.5 \times 10^{-3}$ m) for various initial matric of the dry zone ($\psi_2(t=0)$). Left column (a and b) are for narrow rhizodeposit distributions and right (c and d) are for wide distributions. Top row (a and c) are for sandy loam soil and bottom row (b and d) is for loam soil.

finer textured soils, and hence HR becomes more inhibited due to their lower unsaturated hydraulic conductivity, compared to latter ones. Previous evidence indicated that HR was suppressed and minimal in sandy soils (Burgess, 2000; Wang et al., 2009; Scholz et al., 2008; Prieto et al., 2010) . Elevated water retention by rhizodeposits enhanced unsaturated hydraulic conductivity

more effectively in coarser than finer textured ones. Thus, soil with coarser texture may provide more opportunity for plants to benefit from rhizodeposits and its facilitation in HR wetting.

The simulated results indicate that HR magnitude varied across time and space, e.g., their location at the radial distance from the root surfaces. For example, in this study, HR magnitude was found higher at the close proximity of root surfaces than regions farther away. Note that many other factors may affect HR magnitude. Important factors could beyond the factors that

we illustrated above, such as root length and conductance (Neumann and Cardon, 2012). Therefore, mechanistic understanding must be based on the representation of HR magnitude at the proper scale of interests, and caution must be taken when interpret changes in HR magnitude, given the higher variability in HR magnitude we've observed (Neumann and Cardon, 2012).



## 4.2 The Benefits of Rhizodeposits and HR on Nutrient Uptake and Mineralization

The simulated results indicate the benefits of HR on increased nutrient availability to plants in dry shallow layers, rather than
increased water uptake. The increase in HR magnitude in Zone 2 enabled higher water uptake, however, such little water
uptake in the shallow dry layer, i.e., Zone 2, cannot fulfill the water demands but most of the transpiration came from the
deeper layer, i.e., Zone 1, where soil water was sufficiently available. In contrast, significant increases in net nutrient uptake
and mineralization (0 to up to 150 % and 20 to 60 %) than net water uptake (0 to up to 6 %) were observed for all the scenarios
we simulated, even under scenarios with distinguished soil texture and wetness. These results indicate that the benefits of HR
for increased nutrient availability may outcompete the benefits for water uptake, as hypothesized by Ryel et al. (2002). The
concurrence of higher HR magnitude with localized nutrient distribution in the dry soil patches thus points to the plant-regulated
function of HR in nutrient forging (Prieto et al., 2012b). Indeed, recent experimental evidence suggested that plants initialize
HR to match their nutrient demands rather than water supply (Yan et al., 2020). Here, our simulated results quantitatively
identified such effects to further supports this hypothesis.

The simulated benefits of rhizodeposits in enhanced HR and nutrient availability suggested that plants may utilize rhizode-
posits to amend soil that facilitates HR for effective nutrient extraction. Qualitative evidence of Nambiar (1976) showed that
HR-mediated nutrient uptake may have been facilitated by the formation of rhizoshealth, a typical indicator of the presence
of rhizodeposits, within dry but nutrient-rich soils. Higher amounts of rhizodeposits may thus provide more opportunity for
increased nutrient availability in shallow dry nutrient-rich layers.

Increased rhizodeposits enable higher nutrient availability to plants through two major mechanisms: (1) when the soil dries,
rhizodeposits increase soil water content and connectivity that maintains bidirectional wetting and drying at a high rate, and
therefore, increases active nutrient uptake through effective convective and diffusive influx, and (2) increased soil water content
by rhizodeposits support the microbial activities that favor nutrient mineralization.

Multiple studies support the earlier mechanism, as evidenced by rhizodeposits having a higher water-holding capacity that
increases soil water content, especially under low water potentials (Carminati et al., 2011; Ghezzehei and Albalasmeh, 2015).
The later mechanisms can be supported by findings of a higher decomposition rate of organic compounds by root- and fungus-
induced HR (enhanced carbon mineralization by $2,800\%$ and enzymatic activity by $250 - 350\%$) (Aanderud and Richards,
2009; Guhr et al., 2015). The latter mechanism may not be mutually exclusive from the early one, instead, root release of
water and rhizodeposits through HR in relatively dry soils may create a synergy for the increase in nutrient mineralization.
The higher water content in relatively dryer soils resulting from rhizodeposits alleviates the microbial stress of water shortage.
Besides, the presence of rhizodeposits promotes microsite formation with better accessibility of water, oxygen, and nutrients
for microbial activities. Notice that the solute diffusion exponentially increases with increasing soil water content, resulting in
a higher diffusive nutrient influx at higher levels of rhizodeposits (Zarebanadkouki et al., 2019). Holz et al. (2019) observed
that rhizodeposits increase phosphatase diffusion for organic decomposition that are further away from the root surface. Con-
sequently, those benefits of rhizodeposts creates higher chances of priming for mineralization by selecting microbes that are
more adapted with higher mineralization potentials.





### 4.3 Implications to Plant-Soil Interactions in Non-homogeneous Resource Distribution under Global Climate Change

Past research suggests that HR by roots serves as a critical mechanism of plant adaption to heterogeneous distributions of water and nutrients in the root zone. However, there is still uncertainty on how plants regulate HR occurrence and magnitude. The

need for a steep matric potential gradient to drive HR is well documented (Neumann and Cardon, 2012; Prieto et al., 2012a). However, maintaining HR flux at a relatively high level also requires good root-soil conductance that can withstand rhizosphere drying (Wang et al., 2009; Wang, 2011; Meinzer et al., 2004; Amenu and Kumar, 2008). Our simulated results demonstrated that plants could balance these competing requirements by modifying the rhizosphere hydraulic properties. Rhizodepoists increase hydraulic conductance, hence the volume of HR water, by enhancing soil water retention. Moreover, the declining

slope of rhizodeposit concentration away from root surfaces prevents hydraulic decoupling by avoiding the need for a steep water potential gradient (Carminati et al., 2011; Ghezzehei and Albalasmeh, 2015).

A direct consequence of drying near-surface soils is limiting access to shallow nutrients and slowing down microbial cycling. Therefore, HR can mitigate the effects of such localized dryness and allow plants to thrive by utilizing non-uniformly distributed resources in the soil (Ryel et al., 2010). Our results suggest that the nutrient provisioning service of HR outweighs

that of its water delivery service. We thus postulate that the release of rhizodeposits occurring at the local scale is an orchestrated whole-plant scale response to nutrient demands. This idea of plants utilizing HR as a strategy for nutrient forging is consistent with the observed correlation between nutrient availability and HR magnitude (Prieto et al., 2012a; Yan et al., 2020). In addition, nutrients in the shallow dry layers are often associated with soil organic matter, requiring nutrient mineralization before plants can access the resources. Therefore, HR plays a direct role in unlocking nutrients by stimulating microbial ac-

tivity. This environmental facilitation of microbial activity is distinct from the well-studied priming of nutrient cycling via the exudation of energy-rich organic sugars and acids by roots (Kuzyakov, 2002; Keiluweit et al., 2015). This conclusion is consistent with observed correlations between HR and soil organic matter decomposition (Aanderud and Richards, 2009; Guhr et al., 2015; Armas et al., 2012).

The above conclusions also provide new insights into how plants may adapt to shifts in resource distributions due to climate

change. For example, one of the projected consequences of climate change is prolonged warming that desiccates fertile shallow soil layers (Berg et al., 2017). This forces plants to rely on water storage in deeper subsoil/bedrock (McCormick et al., 2021), and intensifies vertical decoupling of water and nutrient availability across the soil profiles, potentially reducing nutrient uptake and carbon assimilation (Querejeta et al., 2021). Our modeling study suggests that plants that can respond by releasing higher rhizodeposits could adapt more efficiently by maintaining access to shallow nutrients and organic matter deposits. Therefore,

the ability to tune the quality and quantity of rhizodeposit exudation appears to be a crucial trait of phenotypic plasticity needed to adapt to a changing soil environment. Furthermore, selective interaction with mycorrhizal fungi (Allen, 2007) that support HR could play a critical role in ecosystem response to persistent droughts (Williams and de Vries, 2020).



## 4.4  Considerations for Further Research

The composition, physical characteristics, and quantity of rhizodeposits vary with plant type, maturity, and degrees of soil
dryness (Kuzyakov and Razavi, 2019). Therefore, the functions of rhizodeposits also vary with the circumstances that promoted the exudation. For example, differences in viscosity and polarity influence the spatial distribution and wettability of the rhizosphere. While this study focused on hydrophilic rhizodeposits that concentrated near the root surface, other circumstances may call for the exudation of hydrophobic compounds that avert water loss from plants and prevent shock upon rapid wetting. Therefore, systematic cataloging of the nature of rhizodeposits, the conditions for their release, and the services they render
are needed.

This study provided a plausible rationale for short- and long-range feedback that allows plants to forage and utilize resources distributed unevenly in time and space. Furthermore, it suggested that this feedback could be an essential deciding factor in plant adaptation to new environments created by climate change. Our results indicate that answering questions about plant adaptation to complex and changing soil and environmental conditions requires integrating biotic and abiotic feedback in the
soil-plant-atmosphere continuum. For example, the role of soil texture on rhizosphere conductance and HR are well-represented in both experimental and modeling studies (Wang et al., 2009; Wang, 2011; Meinzer et al., 2004; Amenu and Kumar, 2008). Moreover, recent modeling advances have accounted for the effect of HR on biogeochemical cycles (Roque-Malo et al., 2020; Quijano et al., 2013). Our study adds a novel framework for integrating short-distance processes (rhizodeposition-facilitated rhizosphere hydrodynamics) with long-distance feedback (HR). While this model does not aim to describe complete soil
and plant functions, it provides a tool for exploring how plants engineer their immediate surroundings to facilitate resource acquisition.

## 5  Conclusions

Our simulated results demonstrate that alteration of rhizosphere soil by rhizodeposition facilitates HR. Moreover, we show that the magnitude of HR has a positive influence on the active uptake of plant-available nutrients and the rate of organic matter
mineralization. Mechanistically, higher HR magnitude results from the presence of rhizodeposits that increases soil water retention and compensates for the loss of hydraulic conductivity. The facilitation of HR by rhizodeposits enhances diffusion and bulk flow of nutrients and promotes nutrient mineralization in shallow dry nutrient-rich layers. The positive effects of increased rhizodeposits on nutrient uptake and mineralization appears to be more effective in coarser textured soils than finer ones. The investment in rhizodeposits thus seems critical for plants in arid and semi-arid regions that experience frequent
drying of surface soils where nutrients are disproportionately concentrated. Facilitation of HR by rhizodeposition is thus a critical adaption mechanism to buffer the negative impacts of climate warming and drying on nutrient availability in the fertile dry surface layers, which request further research attention.

*Code and data availability.*  Code and data used will be published online. A temporary access is provided for the review





## Appendix A:  Evaluation of Mean Rhizodeposition Concentration

The integral in Eq. (11) was evaluated using MATHEMATICA (Wolfram).

$$\int_{R}^{\infty} X\,dr = \frac{1}{\beta}2\pi X_0 \xi \left( \beta R \log\left(\frac{\xi}{\xi-1}\right) - \text{Li}_2\left(\frac{1}{1-\xi}\right) \right) \tag{A1}$$

where $\text{Li}_2(z) = \sum_{k=1}^{\infty} z^k/k^n$ is the poly-logarithm function and $\xi > 1$. In this study we set $\xi = 1.1087655$, which simplifies $\text{Li}_2(1/(1-\xi)) \approx -4$. This value is close Ghezzehei and Albalsmeh (2014) found that the parameter is close to unity.

Table A1: Definition of symbols and constants

| Symbol | Unit | Constant | Definition (citation) |
|---|---|---|---|
| $a$ | – | -1113.1 | A constant for modification of $\alpha$ by rhizodeposits (Ghezzehei and Albalasmeh, 2015) |
| $A$ | m$^3$ | | Specific-surface area of the soil-root interface |
| $b$ | – | 100 | A prescribed constant describing shape of diurnal boundary condition function |
| $C$ | – | | Dimensionless organic C concentration |
| $D_0$ | m$^2$s$^{-1}$ | $4 \times 10^{-9}$ | Diffusivity of the nutrient in free water (Espeleta et al., 2017) |
| $D_e$ | m$^2$s$^{-1}$ | | Effective dispression-diffusion coefficient |
| $F_{max}$ | m s$^{-1}$ | $3.782 \times 10^{-7}$ | Maximum nutrient uptake rate (Espeleta et al., 2017) |
| $h$ | – | | Exponent of the Hill function, for active nutrient uptake |
| $k$ | $s^{-1}$ | | Effective mineralization rate |
| $k_o$ | $s^{-1}$ | | Effective mineralization rate at optimal physical environmental condition |
| $k_w$ | – | | Dimensionless sensitivity function to moisture and matric head |
| $K$ | m s$^{-1}$ | | Soil hydraulic conductivity |
| $K_N$ | - | $7.882 \times 10^{-1}$ | Dimensionless half-saturation concentration of nutrients (Espeleta et al., 2017) |
| $K_S$ | m s$^{-1}$ | | Saturated hydraulic conductivity |
| $m$ | – | | Shape parameter of van Genuchten soil water retention curve |
| $n$ | – | | Shape parameter of van Genuchten soil water retention curve |
| $N$ | – | | Dimensionless nutrient concentration, defined on per unit soil volume basis |
| $N'$ | – | | Dimensionless nutrient concentration, defined on per unit pore water volume basis |
| $N_l$ | mol m$^{-3}$ | | Nutrient concentration, defined on per unit soil volume basis |
| $N_{\max}$ | mol m$^{-3}$ | | Maximum nutrient concentration, defined on per unit pore water volume basis |
| $r$ | m | | Independent spatial variable, radius within the model domain |
| $R$ | m | | Root radius |
| $R_R$ | m | | Zone of rhizodeposition, $\approx 10R$ |



| $R_\infty$ | m | | Outer boundary of the model domain, determined by $\rho_L$ |
| $R_*$ | m | | Upper limit for integration of soil volume |
| $s$ | m$^{-1}$ | $2.1 \times 10^{-3}$ | Empirical coefficient of moisture sensitivity function $k_w$ (Ghezzehei et al., 2019) |
| $t$ | s | | Independent time variable |
| $t'$ | s | | The duration overwhich short-term cumulative mineralization is computed. |
| $T$ | m$^3$s$^{-1}$ | | Transpiration water flux |
| $X_0$ | kg kg$^{-1}$ | | Mass fraction of the rhizodeposits at the root-soil interface |
| $X$ | kg kg$^{-1}$ | | Mass fraction of the rhizodeposit at $r > R$ |
| $\alpha$ | m$^{-1}$ | | Shape parameter of van Genuchten soil water retention curve |
| $\alpha'$ | m$^{-1}$ | | Shape parameter of van Genuchten soil water retention curve modified by rhizodeposits |
| $\beta$ | – | | Shape factor of rhizodeposit distribution |
| $\theta$ | m$^3$m$^{-3}$ | | Volumetric soil water content |
| $\theta_s$ | m$^3$m$^{-3}$ | | Saturated volumetric soil water content |
| $\theta_r$ | m$^3$m$^{-3}$ | | Residual volumetric soil water content |
| $\Theta$ | m$^3$m$^{-3}$ | | Dimensionless soil saturation |
| $\kappa$ | m$^2$ s$^{-1}$ | $7.882 \times 10^{-1}$ | Conductance of the root-stem-leaf continuum (Sperry et al., 1998) |
| $\lambda$ | m | $10^{-6}$ | Dispersivity of nutrient (**?**) |
| $\xi$ | – | | Shape factor of spatial distribution of the rhizodeposits |
| $\psi$ | $m$ | | Soil matric head |
| $\rho_R$ | m m$^{-3}$ | $7.882 \times 10^{-1}$ | Root length density (prescribed value) |
| $\tau$ | d | | Time described in days |
| $\omega$ | – | | Smoothed step function for switching transpiration on and off |

*Author contributions.* JY and TAG jointly conceptualized the modeling framework and co-authored the underlying code. JY and TAG wrote
the the manuscript collaboratively.

*Competing interests.* We do not have any competing interests.





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
