# Peer review of "Roots induce hydraulic redistribution to promote nutrient uptake and nutrient cycling in nutrient-rich but dry near-surface layers"

_Biogeosciences, 2022_

## Author Comment (AC1)

Response to Reviewer Comments #1

June 15, 2022

**General Comment**

The manuscript by Yan and Ghezzehei presents a soil-plant water transport model including a two layer-based hydraulic redistribution (HR) mechanism. The model is used to study how a prescribed rhizosphere volume around the roots mediates HR effects, with a particularly strong effect on nutrient transport and uptake. The authors also show that the enhanced water content in the rhizosphere can promote decomposition of native organic matter. The topic is interesting and suitable for Biogeosciences, but the manuscript in my view is not always clear and would benefit from a thorough proof-reading. Most important, the proposed model now contains three components—rhizodeposits, organic matter, nutrients—that are connected conceptually but not mathematically. This makes hydrology and biogeochemistry in the model too un-coupled to support the conclusion that rhizodeposits are an adaptive response of plants to face dry conditions. My main comments are explained first, followed by other suggestions.

**Comment 1: Nutrients are modelled independently of organic matter mineralization.**

Nutrients are essentially regarded as a passive tracer, transported through the domain and taken up by plants. With higher hydraulic conductivity in the rhizosphere, nutrients are transported more rapidly according to the model, so the result that rhizosphere improves nutrient uptake is expected. Organic carbon decomposition rate is also enhanced thanks to the higher water content in the rhizosphere, which is also an expected result. In other words, both enhanced nutrient uptake and decomposition rates are a direct consequence of the model

construction. This approach is simple and easy to understand, but has two drawbacks: i) nutrients are produced by decomposition of organic matter (now the two processes are independent) and ii) nutrients can be immobilized and re-mineralized by microbes in the rhizosphere (now only passively transported). Without this dynamic links between nutrient availability, organic matter and rhizosphere environmental conditions, it is difficult to say if nutrient availability actually improves thanks to the rhizosphere environment and HR.

**Response 1**

We thank the reviewer for the thorough and critical review. The other reviewers also had similar criticisms about the conclusion and proofreading of the content. In response, we have made a substantial revision to the content. First, we revised the introduction by adding more background information about the challenges in quantitatively identifying and connecting the short-range and long-range functions of hydraulic redistribution. As noted by the reviewer, physical, biological, and chemical processes in the rhizosphere are complex and interactive. In principle, developing a model that accounts for these interactions is feasible. However, the interpretation will become more complex and intractable as the number of parameters and degrees of freedom increase. Our primary aim was to test our hypothesis that HR is triggered to increase nutrient availability via two possible mechanisms: (a) faster active uptake and (b) faster mineralization. Experimental testing of this hypothesis is very challenging. Therefore, we designed our model to address this difficulty specifically. Separating the effects of HR on transport and mineralization was a necessary condition for this purpose. Moreover, we chose to conduct only short-term simulations to avoid the impact of interactions between mineralization, sorption/desorption, and transport on net uptake (see the response to other questions below for details). We will revise the introduction to reflect these ideas.

**Comment 2: Rhizodeposits are not static and are costly for the plant.**

One component of the model deals with organic matter decomposition, referring to "rhizospheric C mineralization" (L171)—yet, the rhizosphere is regarded as fixed in terms of both extent and hydraulic properties (driven by rhizodeposit mass of C content). However, one could argue that the rhizosphere function needs to be 'maintained' by secreting C as the rhizosphere compounds are mineralized. In the current model there is no connection between rhizosphere and organic matter decomposition, which might be a reasonable approximation given the short time scales of the investigation (individual dry-down period). However, by neglecting the dynamics of rhizosphere properties, it is not possible to estimate the costs for the plant, and thus it is difficult to conclude that rhizodeposits are adaptive and promote plant survival or growth. Is the extra nutrient uptake worth the cost of the rhizodeposits and their maintenance in the long term? Extending the model to study these long-term effects (even in a highly

idealized way) is in my view necessary to "shed light on how plants can adapt to non-ideal resource distribution" (L50) and to support the conclusion that "the investment in rhizodeposits thus seems critical. . . " (L379). If these extensions are not feasible, the scope of the work should be adapted to avoid speculation on long-term adaptations.

**Response 2**

We agree with the reviewer on the dynamic interactions that characterize the rhizosphere. In principle, it is possible to make the secretion of rhizodeposits time-dependent. Moreover, it is also possible to account for the depletion of the rhizodeposits and their effect on hydraulic properties. However, this necessitates the introduction of numerous parameters. Considering the effects of each parameter and their interactions would be very complex if not completely intractable. We believe that limiting the number of independent variables and interactions allows us to answer targeted "all else being equal, what is the effect of rhizodeposit concentration on" type of questions. To illustrate our rationale, let us consider all the possible roles of HR in plant function: (a) no positive effect, (b) aids total water uptake, (c) aids nutrient uptake, and (d) aids both. Our model suggests that aid for water uptake is marginal, while support for nutrient uptake is very significant. Had we considered the depletion of rhizodeposits, the effect would have been smaller (depending on the mineralization rate). Nevertheless, it would not substantially alter the conclusion. Note that by considering a wide range of rhizodeposit concentrations, we already demonstrated the positive correlation between rhizodeposit concentration and HR as well as HR and nutrient uptake.

**Comment 3: Short time frame.**

Linked to the point above, the model is used to simulate a few days during a dry period, which is not enough to balance gains and costs of rhizodeposits. I wonder if at least longer simulations could be attempted to assess the behavior of the model when the soil nears the wilting point. At that point cavitation in the xylem might affect the results (see comment below). Also, longer simulations would allow studying how the deep soil water depletion affects HR occurrence.

**Response 3**

The central assumption behind the above considerations is that the concentration of rhizodeposits does not change within the timeframe of the simulations. Therefore, we selected the short duration for the simulations to match this assumption. In addition to rhizodeposit concentration, a shorter time scale also avoids other factors that could change at more extended time frames but are treated as constants in the model, including root size and age, plant tissue conductance, rhizodeposit composition, and rhizodeposit hydraulic characteristics. As stated above, our primary goal is not to represent the full set of interactions

in the rhizosphere but to test the above-stated hypotheses. We believe, the best compromise, in this case, is to keep all extraneous variables constant and ask how the magnitude of rhizodeposition governs HR, net water uptake, nutrient and mineralization rate. In the revised manuscript, we include all statements that clarify these assumptions and explain clearly the rationale for the model architecture we used.

**Specific Comments**

Below we provide answers to the specific questions and edits that will be incorporated in the revised version.

**General**: the manuscript needs proof-reading and a thorough check of sentence structure (see some examples below)

> **Response**: The manuscript will be carefully revised and follow the the reviewer's suggestions. We will use Grammarly and third-person proof readers to catch for typos and grammatical errors..

**Notation**: I find the notation for the soil layers and leaves counterintuitive—the deeper layer is numbered 1, then moving up we find layer 2, and finally the leaves identified by subscript 0. This numbering makes it hard to follow the text and understand the figures

> **Response**: Thank you for the suggestion. We changed the notation of zone numbers to "0", "1" and "2", to represent leaf surfaces, shallow and deep layers.

**L. 7:** "happy accident" could be re-phrased in more scientific terms

> **Response**: We changed the sentence, and it now appears as "...However, how plant roots regulate hydraulic redistribution for resource scavenging through rhizosphere modification remains unclear...".

**L. 17**: not clear what "roots faced with nutrient and organic matter accumulation" means

> **Response**: We changed the sentence to "...a hypothesis that roots in nutrient-rich but dry shallow soils facilitate hydraulic redistribution via rhizodeposition...".

**L. 19**: missing word "could play"

> **Response**: Added.

**L. 37**: type "Therefore"

> **Response**: Corrected.

**L. 46**: nitrogen is to some degree always present—do you mean that HR increased in nutrient amended plots?

> **Response**: Yes, the reviewer was correct. We changed it to "... HR performed by Sagebrush and Boiss shrubs increased in nutrient amended regions where additional nitrogen sources were supplied...".

**L. 50**: explain/change term "variable resourced"

> **Response**: We removed it and changed the sentence to "...between roots that inhabit regions with heterogeneous distribution of water and nutrients...".

**Model concept**: the model separates roots in two soil layers, defining them in terms of length or area per unit soil volume, but this does not imply that the model describes two long roots (Fig. 1 caption, and in L63); in fact, long roots would imply high hydraulic conductance, which has a series of consequences. I would reformulate in the caption and redraw the figure to avoid mis-interpretation; the model deals with many roots of different lengths, whose properties are defined on a per unit soil volume

> **Response**: The reviewer was correct, while the two "big" root sections were conceptualized, they represent a bunch of roots derived from parameters of root length density within a confined soil volume. Such a modeling approach was not uncommon as documented in [1], where they modeled HR through "big" root concepts. We thus revised the methodology and caption to emphasize that "big" roots represent a bunch of roots within a confined soil volume.

**L. 110**: what are the rhizodeposits made of? Units indicate mass, but in terms of dry/wet weight, mass of C?

> **Response**: In [4], the dry mass fraction of rhizodeposits in soils was derived or parameterized from the previous independent works that characterized the root exudates across a radial distance from the root surfaces [3, 6]. The soluble organic matter and organic acids, i.e., citric, malic, and oxalic acid, were used as representative root exudates to quantify the distribution of rhizodeposits or root exudes. The unit is thus expressed as a dry mass of organic matter per unit of dry soil. To clarify that, we specified the definition in the revision.

> The new paragraph at L. 100 now appears as "The spatial distribution of the rhizodeposit mass fraction, the dry mass fraction of rhizodeposits in a unit mass of dry soils, was parameterized from the previous independent works that characterized the root exudates, e.g., citric, malic, and oxalic acid, across a radial distance from the root surfaces [3, 6]. More detailed derivation and parameterization were described by [4]".

**L. 118**: not clear what "excluding a rhizodeposit free control" means; no rhizodeposit could mean $X_0 = 0$

> **Response**: Yes, the reviewer was correct. We added descriptions of rhizodeposit free control, which are the plants under conditions without additional rhizodeposits, and thus specified using "rhizodeposit free control $(X_0 = 0)$".

**L. 140**: symbol $R_b$ is not defined and not listed in Table A1

> **Response**: We changed $R_b$ to $R_\infty$.

**Eq. 15**: the soil-to-leaf conductance depends on xylem water potential (cavitation curve); that is the essence of the model by Sperry et al. (1998) cited here, but this feature is not included. Given the short duration of the simulations it should not affect the results, but with longer dry-downs it could change the speed of water depletion. Small detail: I would use brackets in the order $[(\dots)]$

> **Response**: We changed the order of brackets in the equation as the reviewer suggested. In the revision, we specified the assumption of constant effective conductance of plant tissue. Also, see our replies to the main comments on the model structure.

> The new paragraph on L. 140-150, including the corrected equation now appears as "

$$f_1|_{r=R_\infty} = (\psi_1 - \psi_0)\frac{\kappa}{A} \qquad (1)$$

$$f_2|_{r=R_\infty} = [\psi_2 - \psi_0 - (1 - w)(\psi_1 - \psi_0)]\frac{\kappa}{A} \qquad (2)$$

> where $\kappa$ is the effective conductance that represents the soil-plant-atmosphere interface [8]. Note that the current model simplifies the system by assuming a constant effective conductance of $\kappa$, given the shorter duration of the simulation. More importantly, it separates the effects of rhizodeposition from the compounding effects of changes in plant tissue conductance, e. g. stem and root xylem conductance. Therefore, this allows quantitatively identifying the impacts of rhizodeposition distribution on HR and its derived benefits".

**L. 150**: why only active uptake? Passive uptake should be relatively easy to include in this framework

> **Response**: The model structure on nutrient uptake in the methodology section was re-clarified by removing the ambiguity.

> The revised paragraph at L. 150-157 now appears as "The nutrient uptake was modeled by considering both passive nutrient flow and

active nutrient uptake to the plant roots. Specifically, the passive nutrient flow from the rhizosphere to the root surface was modeled through convective-dispersive nutrient flux, while active uptake is the result of cation competition on the root surfaces, which was modeled by its relationship to maximum uptake rate and half-saturation concentration, $F_{max}$, and $K_N$ described in [2]".

**L.151**: citation should be formatted as Author (year)

**Response**: Reformatted.

**L.170**: this section is on nutrient mineralization potential according to the title, but describes organic matter decomposition. Nutrient mineralization is related but not quite the same thing

**Response**: See our reply to the question on L. 181.

**L. 181**: how can this model be deemed "effective in describing the role of rhizodeposition in increasing nutrient availability" since it does not describe nutrient mineralization, nor any reaction the nutrient might undergo?

**Response**: Regarding questions to L. 170 and L. 181, the reviewer was correct. Within the revision, we changed nutrient mineralization to organic matter decomposition, which likely improves plants' ability to optimally exploit nutrient distributions in the dry soil regions.

Now the last sentence of the paragraph at L. 181 appears as: "Therefore, this model is deemed effective in describing the role of rhizodeposition in increasing organic matter decomposition/mineralization, which serves as an indicator of plant-available nutrient transformation in the organic-rich but dry surface soil layers".

**L. 188**: extra "s"?

**Response**: Added.

**L.191** why faster water flow? If I understand correctly, water flow is driven by a water potential difference, not water content

**Response**: The reviewer was correct that water flow is driven by water potential difference, but another critical controller is the unsaturated hydraulic conductivity of the rhizosphere soil, which depends on the soil moisture content, as shown in Figure 2b. Essentially, an increase in moisture content leads to a higher rhizosphere soil hydraulic conductivity that enables a faster or higher water flux.

We clarified it by revising the sentence to "resulting in faster water flux due to increased unsaturated soil hydraulic conductivity under wetter conditions".

**L. 201** the label in Fig. 5 indicates average water potential, not water potential in zone 2

> **Response**: We changed the definition in the caption of Figure 5. The new caption now starts with "Temporal variations of mean matric head (volume-weighted mean matric potential in Equation 19b) for one level of initial matric head in zone 2...".

**L.216**: is the nutrient uptake effect noticeable because of no-flow boundary conditions? Nutrients are not produced in this model, just transported and taken up by the plants

> **Response**: Recall that the outer boundary of the model domain is calculated using the root-length density. Therefore 100% of the root zone soil volume is represented in the model. Conceptually, this boundary represents a flux divide (symmetric boundary) between adjacent roots. Because of the radial geometry, the steepest gradient of nutrient concentration is localized near the roots and the concentration at the outer boundary remains largely unaffected. In our simulations, the nutrient uptake was limited to the interior 5 mm, while the outer boundary was set at 17 mm (see Fig 7). The only scenario which results in nutrient depletion at the boundary (hence, the noticeable effect of the no-flux boundary condition) is a long-time simulation with no replenishment. This scenario would be an accurate representation of bulk soil nutrient depletion. Finally, our model captures rhizosphere dynamics at realistic spatial scales: the radius of activity, i.e., 5 mm, was found consistent with results reported in [5], where active nutrient uptake by roots was found around 2-5 mm.

**L. 235 − 246**: multiple typos: space missing, "promotes","pronounced"

> **Response**: Corrected the typos.

**Fig. 9 caption**: the model describes only active nutrient uptake, but the caption mentions passive uptake

> **Response**: We re-clarified the model structure on nutrient uptake in the methodology section (see replies to L. 150). Captions of Figure 9-11 were updated, and now appear as:

> **Figure 9.** Effect of elevated rhizodeposition on relative increase in net water uptake from the effective soil volume occupied by roots ($0.25$ mm $\leq r \leq 17.5$ mm) for various initial matric of the dry zone ($\psi_1(t = 0)$). Left columns (a and b) are for narrow rhizodeposit distributions and right (c and d) are for wide distributions. The top row (a and c) are for sandy loam soil and the bottom row (b and d) is for loam soil.

**Figure 10.** Effect of elevated rhizodeposition on relative increase in net nutrient uptake in the immediate region around roots (0.25 mm $\leq r \leq$ 2.5 mm) for various initial matric of the dry zone ($\psi_1(t = 0)$). Left columns (a and b) are for narrow rhizodeposit distributions and right (c and d) are for wide distributions. The top row (a and c) are for sandy loam soil and the bottom row (b and d) is for loam soil.

**Figure 11.** Effect of elevated rhizodeposition on relative increase in OM mineralization in the immediate region around roots ($0.25 \times 10^{-3}$ m $\leq r \leq 2.5 \times 10^{-3}$ m) for various initial matric of the dry zone ($\psi_1(t = 0)$). Left columns (a and b) are for narrow rhizodeposit distributions and right (c and d) are for wide distributions. The top row (a and c) are for sandy loam soil and the bottom row (b and d) is for loam soil.

**L. 284**: missing verb

**Response**: We changed the sentence to "Importance factors beyond the factors that we illustrated here could also include root length and conductance."

**L.285**: not clear link to previous sentence despite "Therefore"

**Response**: Removed.

**L. 287**: higher variability compared to what?

**Response**: We changed to "large variability in HR magnitude estimated from empirical and modelling studies [7]".

**L. 292**: longer simulations needed (see comment above)

**Response**: Please see our reply to general comments.

**L. 297**: explain/change term "initialize"

**Response**: Changed.

**L. 298**: but if plants are so effective at promoting mineralization of native organic matter, its content will decrease through time, until it will not be able to provide nutrients; an equilibrium will be attained so the relative advantage might be temporary

**Response**: Yes, we agree with the reviewer that the benefits can be temporary, we thus acknowledged the nature of the benefits could be temporal and short-termed in the revision.

**L. 312**: enhanced mineralization with respect to what control/reference state?

**Response**: We revised the sentence by adding "in comparison to controls that HR was inhibited".

**L. 326**: define "good"

**Response**: We changed "good" to "moderate- and high-level of root-soil conductance that can withstand rhizosphere drying without limiting bidirectional water flow".

**L. 332**: previously it was argued that microbial activity was stimulated, not slowed down

**Response**: Sentence removed.

**L. 338**: why only "often" and not always? are you referring to fertilized systems?

**Response**: Yes, we referred to other contrasting systems with additional inorganic fertilization. We thus modified the sentence by specifying "...in natural systems with limited fertilization of inorganic chemicals...".

**Table A1**: check units of A, now a volume, but labelled as specific surface area

**Response**: Corrected by changing to "$m^2/m^3$".

**References**

[1] G. G. Amenu and P. Kumar. A model for hydraulic redistribution incorporating coupled soil-root moisture transport. *Hydrol. Earth Syst. Sci.*, 12 (1):55–74, Jan. 2008. ISSN 1607-7938. doi: 10.5194/hess-12-55-2008. URL https://www.hydrol-earth-syst-sci.net/12/55/2008/.

[2] J. F. Espeleta, Z. G. Cardon, K. U. Mayer, and R. B. Neumann. Diel plant water use and competitive soil cation exchange interact to enhance NH4+ and K+ availability in the rhizosphere. *Plant and Soil*, 414(1):33–51, May 2017. ISSN 1573-5036. doi: 10.1007/s11104-016-3089-5. URL https://doi.org/10.1007/s11104-016-3089-5.

[3] Y. Gao, Y. Yang, W. Ling, H. Kong, and X. Zhu. Gradient Distribution of Root Exudates and Polycyclic Aromatic Hydrocarbons in Rhizosphere Soil. *Soil Science Society of America Journal*, 75(5): 1694, 2011. ISSN 0361-5995. doi: 10.2136/sssaj2010.0244. URL https://www.soils.org/publications/sssaj/abstracts/75/5/1694.

[4] T. A. Ghezzehei and A. A. Albalasmeh. Spatial distribution of rhizodeposits provides built-in water potential gradient in the rhizosphere. *Ecological Modelling*, 298:53–63, Feb. 2015. ISSN 03043800. doi: 10.1016/j.ecolmodel.2014.10.028. URL https://linkinghub.elsevier.com/retrieve/pii/S0304380014005262.

[5] Y. Kuzyakov and B. S. Razavi. Rhizosphere size and shape: Temporal dynamics and spatial stationarity. *Soil Biology and Biochemistry*, 135:343–360, Aug. 2019. ISSN 0038-0717. doi: 10.1016/j.soilbio.2019.05.011. URL `http://www.sciencedirect.com/science/article/pii/S0038071719301452`.

[6] M. Li, T. Shinano, and T. Tadano. Distribution of exudates of lupin roots in the rhizosphere under phosphorus deficient conditions. *Soil Science and Plant Nutrition*, 43(1):237–245, Mar. 1997. ISSN 0038-0768, 1747-0765. doi: 10.1080/00380768.1997.10414731. URL `http://www.tandfonline.com/doi/abs/10.1080/00380768.1997.10414731`.

[7] R. B. Neumann and Z. G. Cardon. The magnitude of hydraulic redistribution by plant roots: a review and synthesis of empirical and modeling studies: Tansley review. *New Phytologist*, 194(2):337–352, Apr. 2012. ISSN 0028646X. doi: 10.1111/j.1469-8137.2012.04088.x. URL `http://doi.wiley.com/10.1111/j.1469-8137.2012.04088.x`.

[8] J. S. Sperry, F. R. Adler, G. S. Campbell, and J. P. Comstock. Limitation of plant water use by rhizosphere and xylem conductance: results from a model. *Plant, Cell & Environment*, 21(4):347–359, 1998. ISSN 1365-3040. doi: 10.1046/j.1365-3040.1998.00287.x. URL `https://onlinelibrary.wiley.com/doi/abs/10.1046/j.1365-3040.1998.00287.x`.

---

## Author Comment (AC2)

Response to Reviewer Comments #2

June 15, 2022

**General Comment**

In the paper entitled "Roots induce hydraulic redistribution to promote nutrient uptake and nutrient cycling in nutrient-rich but dry near-surface layers", the authors present a mathematical model of soil hydraulics, aimed to investigate the effects of soil modification by rhizosphere deposition on water and nutrient uptake. I think the mathematical model and the analyses presented in this paper are sound and of scientific value, but I do have two main issues with the presentation of the work. First, I think the framing of the work is not consistent with the methodology, and should be reconsidered. Second, I think that the writing is not up to the standard I expect for publication, and that the paper requires a thorough proof read to improve the text.

   The authors present the following knowledge gap in the abstract of the paper: "whether hydraulic redistribution is a passive happy accident or a process controlled by plants remains unclear". In the introduction, the authors state: "The exact mechanism by which roots can induce HR is, however, not known. Here, we present a modelling study that demonstrates that alteration of rhizosphere soil by rhizodeposition facilitates HR". I am of the opinion that this knowledge gap is not one that can be filled with the mathematical modelling approach presented in this paper. The fact that one can simulate HR by changing rhizosphere properties in a mathematical model does not provide evidence that real plants can do the same, and it will this not shed light on the mechanisms by which roots can induce HR. I am of the opinion that the paper should be reframed to focus on the results that show how changes in rhizosphere properties differentially affect water uptake, nutrient uptake, and nutrient mineralisation in different soils and soil water conditions. This can then be framed as a series of hypotheses that should be tested with experiments: namely that plants

are able to modify soil properties in the way that was tested with the model through rhizosphere depositions, and that the exudation of these deposits is more common or more pronounced in soil types and soil water conditions where the model predicts the largest increase in the uptake of either water (under wet conditions and in sandy loams) or nutrients (under dry conditions and in sandy loam). However, I think that the authors currently overextend the impact of this work in sections 4.3 and 4.4. I think framing this work in the context of climate change is not appropriate in the paper's current form, and a statement such as the one made in L. 363-365 ("Our results indicate that answering questions about plant adaptation to complex and changing soil and environmental conditions requires integrating biotic and abiotic feedback in the soil-plant-atmosphere continuum") does not match up with the model presented in this paper. Either the modelling should be extended significantly to include more than just a hydraulic model, or the framing of this work should not go beyond the metrics simulated by the model (i.e. water and nutrient uptake). This reframing should lead to significant re-writing of the introduction and the discussion (sections 4.3 and 4.4 in particular).

> **Response**: We thank the reviewer for the constructive criticisms and suggestions. We agree that the presentation of the model, its design philosophy, and what it can and cannot answer were not presented clearly. The revised manuscript, including the introduction and discussion will be much clearer and take into consideration the above criticisms.
>
> The main motivation for this modeling work was to provide a mechanistic explanation for (a) whether roots can actively regulate HR via exudation of rhizodeposits and (b) whether the primary function of HR is for water uptake or nutrient uptake. A significant amount of work done on HR has focused on the water-stress benefit of HR, including a modeling study co-authored by one of us [2] and the references cited therein. Although there are not that many studies that focused on the nutrient cycle/uptake function of HR, an elegant field study by Cardon and co-workers [1] showed evidence for enhanced surface soil nitrogen cycling and nitrogen uptake under HR in sagebrush.
>
> In this study, we set out to develop a physics-based model that retains what we deemed only the essential components of the soil-root-plant system to explore cause and effect relationships that are difficult to conduct in experimental settings. We agree with the reviewer that the model does necessarily reflect the full plant behavior. However, provided that the assumptions and parameterization of a model are consistent with the current knowledge, we believe models (including ours) are effective tools for providing explanations that are direct outcomes of the current understanding. Specifically, the conclusions that we presented in this paper are direct consequences

of the rationale and assumptions that we used in constructing the model.

As simple as it is, our model captured the complex feedback by which plants can adapt to heterogeneous resource distributions in soil profile. In the context of global change modeling, this model provide a more nuanced explanation of the potential resilience than what is represented by modeling vegetation migration/adaptation models that rely primarily on correlations of abundance with mean precipitation and temperature.

In the revised manuscript, we will provide clear distinctions between conclusions that can be reached from this work alone and open questions and suggestions for future considerations.

I would suggest that the paper is carefully proofread to improve the text, which is currently full of grammatical flaws, especially in the introduction and discussion sections. I will mention a few examples I found in the first couple of paragraphs, but have not put the entire text to this level of scrutiny.

**Response**: We corrected the issues listed in the examples and thoroughly reviewed the manuscripts to correct the grammatical flaws.

**Specific Comments**

**L. 27**: I guess this is the explanation of hydraulic redistribution? Please cue the term here if that is the case.

**Response**: Yes, we added the definition, "previously defined as hydraulic redistribution (HR)".

**L. 28**: The current sentence structure reads as if roots modify their immediate surroundings and also modify hydraulic redistribution. Change the order of these two statements to correctly follow up on ". . . separate advances in our understanding of. . . '

**Response**: We modified the sentence to "This study builds upon separate advances in our understanding of how roots modify their immediate surrounding to facilitate HR and HR-driven benefits".

**L. 29**: Replace '-an' with ', which is', and follow this with a better definition of the rhizosphere.

**Response**: We changed to sentence to "...the rhizosphere, which is a narrow region of soil in direct proximity to root surfaces where roots and soils interact".

**L. 32**: Rewrite by switching the order of plant water uptake and wetness of the rhizosphere.

**Response**: We changed the sentence by switching the orders of those phrases.

**L. 33**: 'the rhizosphere's carbon investment' suggests that it is the rhizosphere that is doing the investing.

**Response**: We changed the sentence to "...plant roots' carbon investment in the rhizosphere...".

**L. 36**: Change 'peculiar' to 'specific'

**Response**: Changed.

**Specific Comment**: Finally, I have a couple of minor comments.
In the current analysis, do the results of changes in nutrient uptake include the combined effect of increased nutrient uptake and increased nutrient mineralisation? If so, it would be insightful to decouple the two, presenting both their individual effects and their combined effect.

**Response**: Yes, it's decoupled. We re-clarified that in the introduction and methodology by specifying our modeling goals and procedures of the decoupling. Please also see our reply to the general comment.

**L. 302**: What is a rhizoshealth? (L. 302) I presume this should read rhizosheath, but I would still like an explanation of the term.

**Response**: Changed to "rhizosheath" and added more details on the definition, which is "...the formation of rhizosheath, a portion of the soil that adheres to the roots upon excavation of the root systems, which is considered as a typical...".

**Specific Comment**: The reasoning behind some of the model's design is explained in detail in the discussion, but I feel this should have been explained in the introduction: L. 305-308 – 'Increased rhizodeposits ... nutrient mineralisation.'

**Response**: We followed the suggestion by adding more details on the modeling goals in the introduction. Please see our reply to the general comments.

**References**

[1] Z. G. Cardon, J. M. Stark, P. M. Herron, and J. A. Rasmussen. Sagebrush carrying out hydraulic lift enhances surface soil nitrogen cycling and nitrogen uptake into inflorescences. *Proceedings of the National Academy of Sciences*, 110(47):18988–18993, Nov. 2013. ISSN 0027-8424, 1091-6490. doi: 10.1073/pnas.1311314110. URL http://www.pnas.org/cgi/doi/10.1073/pnas.1311314110.

[2] A. Carminati, E. Kroener, M. A. Ahmed, M. Zarebanadkouki, M. Holz, and T. Ghezzehei. Water for Carbon, Carbon for Water. *Vadose Zone Journal*, 15(2):1–10, 2016. doi: 10.2136/vzj2015.04.0060. URL http://dx.doi.org/10.2136/vzj2015.04.0060.